# Motif-Aware Embeddings for Enhanced Few-Shot Graph Learning

## Abstract

Graph Neural Networks (GNNs) have shown strong performance in node classification tasks. However, in real-world scenarios, only a limited number of nodes are often labeled, leading to the few-shot node classification problem, which is a significant challenge for GNNs. Most existing research focuses on designing new models to adapt to this setting, but often overlooks structural information in the graph data, such as motif patterns, which can provide crucial cues for learning from a few examples. In this paper, we propose a novel framework that integrates motif representations into graph few-shot learning models. Specifically, we extract unique motif representations from the graph and introduce them as virtual nodes. To capture richer structural patterns, we further enhance motif extraction by adding cluster labels based on node similarity, thereby incorporating both structural and feature information. Additionally, we assign TF-IDF scores as edge weights between virtual motif nodes and original nodes to quantify the importance of their connections. Experimental results demonstrate that our approach consistently improves the performance of various graph few-shot learning methods.

## 1 Introduction

Recently, Graph Neural Networks (GNNs) have achieved remarkable success in graph-related tasks such as node classification and link prediction. However, in real-world scenarios, graphs are often sparsely labeled or lack sufficient annotations, which poses a significant challenge for GNN-based models. This problem setting is commonly referred to as graph few-shot learning. To address this, several graph few-shot learning models have been proposed, demonstrating promising performance in improving GNNs under limited supervision.

Most existing research has primarily focused on improving the model architecture itself to achieve better performance. A common direction is the application of meta-learning frameworks (Kim et al., 2023; Huang & Zitnik, 2021; Wang et al., 2022; 2023a). Some studies further extend this line of work by integrating contrastive learning with meta-learning frameworks (Wang et al., 2023b; Liu et al., 2025; 2024). However, these approaches largely overlook an important aspect: even when label information is limited, the graph structure itself still contains latent relational information, such as node connections and degree distributions. Therefore, it is crucial to exploit such information from the original graph, as it can provide additional support for training graph few-shot learning models.

In NLP tasks, researchers often encode frequently occurring words or subwords as unique embeddings, known as word embeddings (Mikolov et al., 2013). These embeddings can then be leveraged by language model to better understand input sentences. In the field of graph learning, recurring connection patterns in a graph are referred to as motifs (Milo et al., 2002). Inspired by word embeddings, we introduce motif embeddings as an additional component for graph few-shot learning. However, several challenges remain. First, there is currently no framework to effectively integrate motif embeddings with the original graph. Second, when introducing motifs into FSL, it is difficult to control the number of motif patterns or potential extra structure that are incorporated.

To address these limitations, we propose a novel framework called **"MoEFL"**, **Mo**tif-**E**mbedding-**F**ew-shot-**L**earning, that integrates motifs into graph few-shot learning models. Specifically, each extracted motif pattern is treated as a virtual node, which is connected to nodes in the original graph if those nodes appear in a subgraph corresponding to the motif. To enrich the extracted motifs, we

introduce cluster labeling, which assigns temporary labels to each node in the original graph based on feature similarity. These temporary labels serve as additional information when extracting motifs, allowing for richer motif representations and easy control over the number of motifs by adjusting the number of clusters. Furthermore, we assign weights to the virtual edges using a TF-IDF score, quantifying the relevance between nodes and motifs. Overall, our framework provides a flexible and effective way to incorporate structural patterns into graph few-shot learning, improving the model's ability to capture complex graph structures.

## 2 RELATED WORK

### 2.1 GRAPH FEW-SHOT LEARNING

Graph few-shot learning aims to enhance the performance of graph neural networks (GNNs) (Xu et al., 2019) when labeled data is scarce. A common strategy is to integrate GNNs with meta-learning frameworks such as MAML (Finn et al., 2017; Rajeswaran et al., 2019) and ProtoNetSnell et al. (2017). Several studies have explored this integration from different perspectives. For example, G-META (Huang & Zitnik, 2021) trains models on local subgraphs surrounding the target node to extract more relevant information. Tent (Wang et al., 2022) reduces variance across nodes, classes, and tasks, while TEG (Kim et al., 2023) introduces task-equivariant graphs that adapt to transformations and capture transferable patterns. In parallel, contrastive learning has also been employed to boost performance. COSMIC (Wang et al., 2023b) is the first to incorporate contrastive learning for better generalization on novel classes, and STAR (Liu et al., 2025) further refines this approach by introducing optimal transport and set representation.

### 2.2 MOTIF IN GRAPH

A motif is defined as a recurring subgraph pattern that frequently appears in a network (Milo et al., 2002). Motifs have been extensively studied in graph-related research across various domains (Das & Dai, 2007; Abou Assi et al., 2018; Thandapani et al., 2013). Recently, they have also been incorporated into graph learning models. For instance, (Chen et al., 2023) integrate motifs into GNNs to enhance model performance, while (Yu & Gao, 2025) leverage motifs to improve the interpretability of GNN outputs.

## 3 MOTIF FOR GRAPH FEW-SHOT LEARNING

In this section, we introduce our proposed framework, which integrates motif embeddings into few-shot node classification. Embeddings are widely used to represent complex data in a continuous vector space. In NLP, words or subwords are mapped to embeddings that capture underlying meaning, which is know as "word embedding" (Milo et al., 2002). These embeddings enable models to better process text. Similarly, in graphs, motifs are recurring subgraph patterns that encode structural information. Inspired by word embeddings, we aim to leverage motif embeddings to enhance few-shot learning on graphs with a framework named **MoEFL**, which is **Mo**tif-**E**mbedding-**F**ew-shot-**L**earning framework.

Specifically, we extract common motifs from the original graph and represent them as virtual nodes with assigned feature embeddings. These virtual nodes are then incorporated into the learning process alongside the original graph. To further control the number and influence of added structures, we introduce cluster labeling to enrich supervision and a TF-IDF–based weighting to regulate virtual nodes and edges.

### 3.1 PRELIMINARY

Given a graph $G$ that is represented by $\{V, \epsilon\}$, where the $V$ denote the vertex $\{V_1......V_n\}$, and $\epsilon$ denotes the edge $\{\epsilon_1......\epsilon_n\}$. In this paper, we focus on the few-shot node classification problem, where only a limited number of node labels are available during the training stage. To enable effective model learning under this constraint, a general way is to adopt a meta-learning framework that constructs a large number of tasks. Each task is defined by $N$ classes, with each class containing $K$ nodes, following the standard "N-way-K-shot" formulation. During the test phase, evaluation is

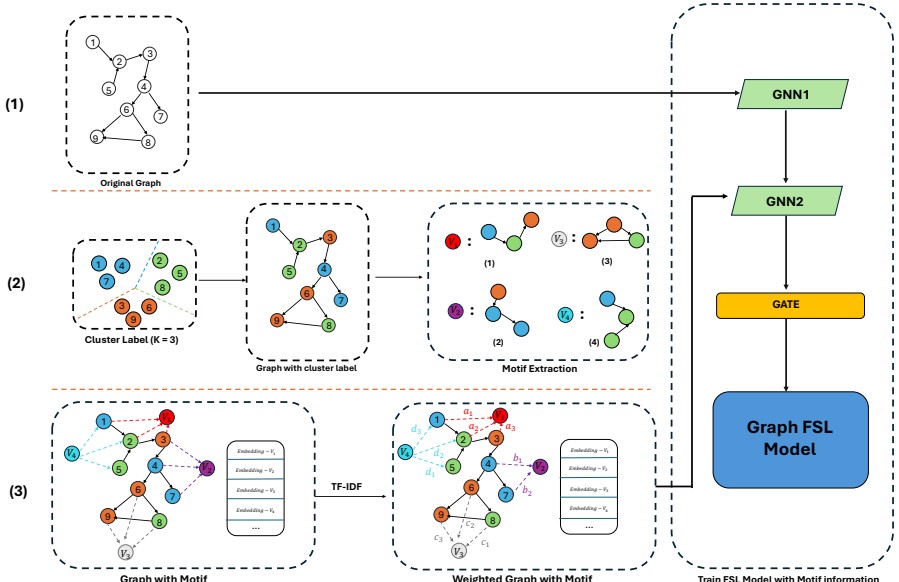

Figure 1: The illustration of our proposed framework. (1):We start from the original graph. (2) based on node feature similarity and assign temporary labels andextract motifs using these labels. (3): introduce virtual nodes, with virtual edge weights determined by TF–IDF (low-score edges are filtered); apply a two-layer GNN to combine original node features and motif embeddings, with a gating mechanism to control the features passed to the few-shot learning model.

conducted on tasks with the same " N-way-K-shot" setting but involving novel classes that were not observed during training.

## 3.2 MOTIF EMBEDDING TO ENHANCE GRAPH FEW-SHOT LEARNING

One of the central challenges in graph few-shot learning is the scarcity of label information within each task, which hinders graph neural networks (GNNs) from effectively learning discriminative node representations. To address this issue, it is crucial to exploit richer sources of structural information from the underlying graph. Motifs, defined as frequently occurring subgraph patterns, capture essential structural relationships that reflect both node properties and feature dependencies. In this work, we propose to incorporate motifs as an additional source of information to enhance model performance in few-shot scenarios.

For a given graph, we follow the approach in (Chen et al., 2023) to extract 3-node motifs, which capture the structural relationships between nodes and all its neighboring nodes.To incorporate the extracted motif within the original graph, we propose to create virtual nodes to represent the motif pattern we extracted. In addition to the original graph $G$, we create $K$ virtual nodes $V^{'}$, which $K$ equals the number of motif extracted from $\mathcal{G}$. Each virtual node represents one motif pattern. Then, each node in the original graph will be connected with the virtual node if they emerge in a sugraph that match with the motif represented by the virtual node. For the embedding of the virutal node created, we apply an learnable embedding $Z$ for each virtual node $V_i^{'}$.

To enable interaction between the embeddings of original nodes and virtual nodes, we employ a two-layer graph neural network (GNN). In the first layer, the original node features are transformed into dense representations based on the initial node features and the adjacency matrix. The second layer takes the output from the first layer and concatenates it with the embeddings of the virtual nodes. At this stage, the adjacency matrix is also updated to contain the connections of the virtual nodes. A final gate will be used to control the feature that actually passed into the FSL model. The

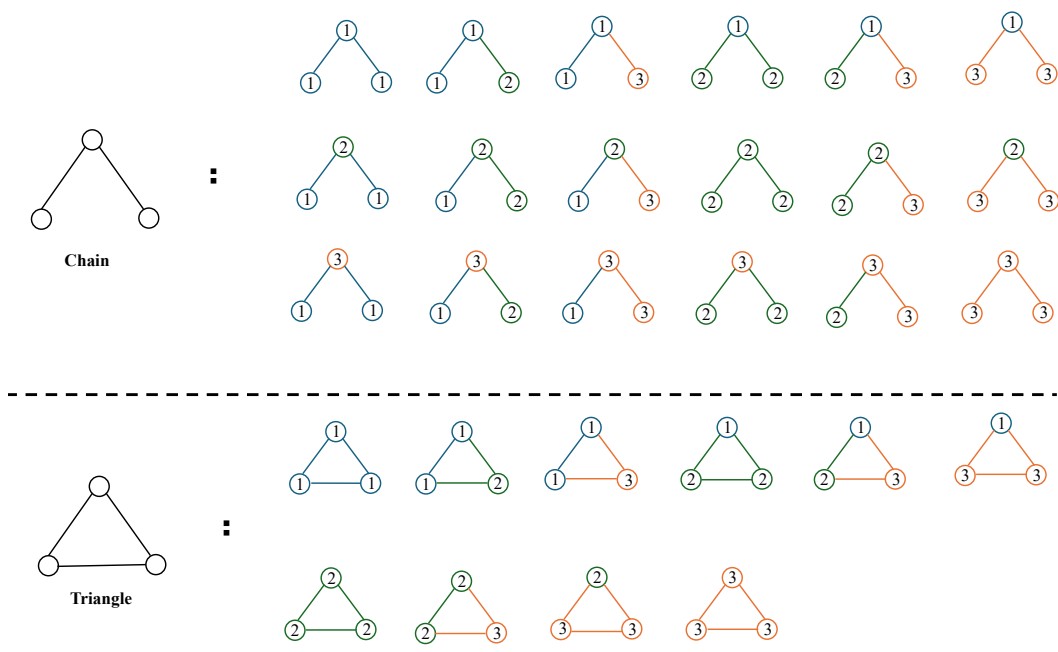

Figure 2: Illustration of motifs with node labels. In an undirected graph without node labels, only two types of 3-node motifs exist: a chain and a triangle. With a cluster label 3, the number of distinguishable motifs increases to 18 for chain-shaped motifs and 10 for triangle-shaped motifs.

overall process can be formalized by the following equations:

$$H_1 = GNN_1(X, A),$$
$$H_2 = GNN_2\Big([H_1 \,\|\, Z], \tilde{A}\Big),$$
$$H_2^{'} = f(\sigma(H_2)) \tag{1}$$

where the $X$ is the original node feature, $A$ is the original adjacency matrix, $Z$ is the learnable embedding for the motif and $\tilde{A}$ is the adjacency matrix with the virtual edge included.

### 3.3 EXPAND MOTIF VIA CLUSTER LABELING

In the previous section, we introduced a framework that incorporates motifs as additional information for graph few-shot learning. However, conventional motif extraction methods rely solely on structural information, which significantly limits the diversity of motif patterns. For example, when considering 3-node motifs, there are only 13 possible patterns in directed graphs, and this number drops to just 2 in undirected graphs. Such a restriction greatly reduces the variety of motifs and consequently hinders the performance of the framework.

To enrich motif diversity without increasing their number, we propose incorporating node information into the motif extraction process. In few-shot learning, access to class labels is limited, so node feature similarity serves as the primary signal for characterizing relationships between nodes. To leverage this signal, we cluster all nodes in the original graph into $N$ groups based solely on feature similarity and assign each node a temporary label according to its cluster membership. These cluster-based labels act as surrogate attributes, enabling the extraction of more diverse and informative motifs.

As illustrated in Figure 2, The use of cluster-based labels greatly increases motif variety. For example, in the case of regular 3-node motifs, clustering the nodes of the original graph into three groups (without considering edge directions) expands the number of motif patterns to 28. These newly extracted motifs capture richer structural information that goes beyond simple connectivity. For instance, the motif $1 \to 2 \to 3$ may represent a very different interaction pattern from $1 \to 3 \to 2$

even though they share the same underlying structure. Traditional motif extraction methods overlook such differences, whereas our approach is able to distinguish them effectively. Importantly, this method does not rely on ground-truth labels and offers flexible control over the granularity of extracted features through the choice of the number of clusters.

### 3.4 Weighted Edge between Motif and Original Node

In last section, we use the cluster label to enrich our motif pattern. However, directly connecting virtual nodes to all associated original nodes can lead to a substantial increase in the number of edges. For instance, if a motif connects 10,000 nodes, its corresponding virtual node would form 10,000 edges, far exceeding the typical degree range of 10 to 20 in the original graph. This imbalance can cause the virtual nodes to dominate the message-passing process during training, potentially overshadowing the original node features and disrupting the overall flow of information in the graph.

To address this issue, we propose assigning additional weights to the edges between original and virtual nodes based on their importance. To evaluate the importance of each edge in a consistent manner, we introduce a variant of the Term Frequency–Inverse Document Frequency (TF-IDF) metric. TF-IDF is widely used in information retrieval to measure the relevance of a word to a document within a corpus. In our setting, we adapt this concept to assess the relevance between an original node and a motif. The adapted definition of TF-IDF in the graph context is described as follows:

*Theorem 1: Term Frequency (TF) in graph is defined as the number of subgraphs in which a given node v participates that are isomorphic to a specific motif m.*

$$TF(v, m) = \log\big(\#\text{Iso}(\text{subgraph}(v), m)\big), \tag{2}$$

where $\#\text{Iso}(\text{subgraph}(v), m)$ denotes the number of isomorphic occurrences between the subgraph centered at node $v$ and the motif $m$.

Where $v$ is a node in the graph, and $m$ is the motif pattern of interest. To control the scale of the "TF" values, a logarithmic transformation is applied. The "TF" term quantifies how strongly motif m is associated with node v.

*Theorem 2: Inverse Document Frequency (IDF) in graph is defined as the number of nodes participate in the motif*

$$IDF(m, v) = \frac{count(m, v)}{count(v)} \tag{3}$$

where v is a node in the graph, and m is the motif pattern of interest. The IDF term reflects the overall relevance of motif m within the graph. The final TF-IDF score will be the product of TF term and IDF term.

After computing the TF-IDF scores, we can filter out certain motifs based on their TF-IDF values. If a motif appears only a few times in the graph—i.e., it is connected to only a small number of nodes—it may not be representative enough to capture meaningful structural patterns. Conversely, if a motif is connected to the majority of nodes, it likely carries little discriminative information. By leveraging the IDF score, we can identify and exclude such motifs from the updated graph. Once the relevant motifs are selected, we compute the TF-IDF score for each edge between a node and a motif, and use this score as the edge weight to reflect its relative importance.

### 3.5 MoEFL Framework

In the previous section, we discussed three components: treating motif patterns as virtual nodes, using cluster labeling to enrich motif patterns, and leveraging TF-IDF scores to weight virtual edges. In this section, we present our framework, illustrated in Figure 1, which can be seamlessly integrated with various existing graph few-shot learning methods.

## 4 Experiment

In this section, we conduct various experiments to prove the effectiveness of our framework.

Table 1: Performance evaluation on **Cora** and Cora-ML datasets. We report average accuracy.

| Dataset | Cora | | Cora-ML | |
|---|---|---|---|---|
| Setting | 2W3S | 2W5S | 2W3S | 2W5S |
| G-Meta (Reg) | 74.39 ± 2.69 | 80.05 ± 1.98 | 88.68 ± 1.36 | 92.16 ± 1.14 |
| TENT (Reg) | 58.25 ± 2.23 | 66.75 ± 2.19 | 55.65 ± 2.19 | 58.30 ± 2.05 |
| X-FNC (Reg) | 78.19 ± 3.25 | 82.70 ± 3.19 | 85.62 ± 3.12 | 90.36 ± 2.99 |
| TLP (Reg) | 71.10 ± 1.66 | 86.15 ± 2.19 | 85.32 ± 2.12 | 58.30 ± 2.05 |
| Meta-GPS (Reg) | 80.29 ± 2.15 | 83.79 ± 2.10 | 87.91 ± 2.12 | 91.66 ± 2.52 |
| COSMIC (Con) | 65.37 ± 2.49 | 69.10 ± 2.30 | 66.02 ± 2.29 | 72.10 ± 2.25 |
| TEG (Reg) | 80.65 ± 1.53 | 84.50 ± 2.01 | 85.10 ± 1.75 | 87.00 ± 1.90 |
| TEG+MoEFL | **84.60 ± 0.11** | **87.79 ± 0.11** | **87.39 ± 0.15** | **92.00 ± 0.08** |
| STAR (Con) | 86.37 ± 2.13 | 88.87 ± 1.68 | 91.20 ± 1.12 | 95.83 ± 0.97 |
| STAR+MoEFL | **88.13 ± 0.04** | **89.90 ± 0.02** | **96.66 ± 0.01** | **96.86 ± 0.01** |

Table 2: Performance evaluation on **WikiCS** and **CiteSeer**. We report average accuracy.

| Dataset | WikiCS | | CiteSeer | |
|---|---|---|---|---|
| Setting | 2W3S | 2W5S | 2W3S | 2W5S |
| G-Meta (Reg) | 61.09 ± 2.84 | 78.35 ± 2.60 | 57.59 ± 2.42 | 62.49 ± 2.30 |
| TENT (Reg) | 68.85 ± 2.42 | 70.35 ± 2.26 | 65.20 ± 3.19 | 67.59 ± 2.95 |
| X-FNC (Reg) | 83.80 ± 3.42 | 86.30 ± 3.20 | 67.96 ± 3.10 | 70.29 ± 3.05 |
| TLP (Reg) | 83.09 ± 2.72 | 70.35 ± 2.26 | 71.10 ± 2.17 | 75.55 ± 2.03 |
| Meta-GPS (Reg) | 85.72 ± 2.10 | 87.05 ± 1.35 | 69.95 ± 2.02 | 72.56 ± 2.06 |
| COSMIC (Con) | 85.51 ± 2.30 | 86.72 ± 1.90 | 70.22 ± 2.56 | 75.10 ± 2.30 |
| TEG (Reg) | 86.20 ± 1.95 | 87.70 ± 2.49 | 73.39 ± 1.59 | 76.69 ± 2.12 |
| TEG+MoEFL | **89.40 ± 0.12** | **92.39 ± 0.10** | **75.99 ± 0.16** | **77.59 ± 0.13** |
| STAR (Con) | 86.40 ± 1.92 | 87.93 ± 0.68 | 76.50 ± 2.12 | 79.43 ± 0.45 |
| STAR+MoEFL | **87.53 ± 0.11** | **88.70 ± 0.10** | **80.20 ± 0.03** | **79.49 ± 0.02** |

## 4.1 DATASET INTRODUCTION

In this work, we choose to work on 6 datasets. Four small datasets and two large datasets. In this section, we will provide some imformation on the dataset we used.

- **Cora** (Yang et al., 2016): A citation network graph. A node represents a document, and an edge represents a citation relationship. It contains **2708** nodes and **10556** edges.

- **CoraML** (Bojchevski & Günnemann, 2018): A citation network where include Machine learning papers. It contains **2295** nodes and **16316** edges.

- **WikiCS** (Mernyei & Cangea, 2022): A graph containing CS articles in Wikipedia. Nodes represent a document, and edges represent hyperlinks. It contains **11701** nodes and **216123** edges.

- **Citeseer** (Yang et al., 2016): A citaion network similar to "Cora".It contains **3327** nodes and **9104** edges.

- **CoraFull** (Bojchevski & Günnemann, 2018): A full citation network graph. It contains It contains **19793** nodes and **65311** edges.

- **CoauthorCS** (Bojchevski & Günnemann, 2018): A graph represent co-authorship. The node represents author and edge represent two author co-author one paper. It contains It contains **18333** nodes and **81894** edges.

## 4.2 BASELINE

For graph few-shot learning, there are two main methodological paradigms: **regular meta-learning** and **meta-learning with contrastive learning**. In this work, we evaluate our method on two state-

Table 3: Performance evaluation on **CoraFull**. We report average accuracy.

| Setting | 5W3S | 5W5S | 10W3S | 10W5S |
|---|---|---|---|---|
| G-Meta (Reg) | $57.52 \pm 2.41$ | $62.43 \pm 3.11$ | $53.92 \pm 2.91$ | $58.10 \pm 3.02$ |
| TENT (Reg) | $64.80 \pm 4.10$ | $69.24 \pm 4.49$ | $51.73 \pm 4.34$ | $56.00 \pm 2.09$ |
| X-FNC (Reg) | $69.32 \pm 3.10$ | $71.26 \pm 4.19$ | $49.63 \pm 4.45$ | $53.00 \pm 3.93$ |
| TLP (Reg) | $66.32 \pm 2.10$ | $71.36 \pm 4.49$ | $51.73 \pm 4.34$ | $56.00 \pm 2.05$ |
| Meta-GPS (Reg) | $65.19 \pm 2.35$ | $69.25 \pm 2.52$ | $61.23 \pm 3.11$ | $64.22 \pm 2.66$ |
| COSMIC (Con) | $73.03 \pm 1.78$ | $77.24 \pm 1.52$ | $65.79 \pm 1.36$ | $70.06 \pm 1.93$ |
| TEG (Reg) | $74.24 \pm 1.03$ | $76.37 \pm 1.92$ | $60.00 \pm 1.16$ | $64.56 \pm 1.04$ |
| TEG+MoEFL | $\mathbf{75.36 \pm 0.096}$ | $\mathbf{81.52 \pm 0.096}$ | $\mathbf{61.87 \pm 0.073}$ | $\mathbf{65.44 \pm 0.082}$ |
| STAR (Con) | $77.77 \pm 0.10$ | $81.24 \pm 0.98$ | $68.60 \pm 0.63$ | $73.53 \pm 0.49$ |
| STAR+MoEFL | $\mathbf{79.52 \pm 0.011}$ | $\mathbf{82.57 \pm 0.016}$ | $\mathbf{69.50 \pm 0.0067}$ | $\mathbf{74.05 \pm 0.0087}$ |

Table 4: Performance evaluation on **Coauthor-CS**. We report average accuracy.

| Setting | 2W3S | 2W5S | 5W3S | 5W5S |
|---|---|---|---|---|
| G-Meta (Reg) | $92.14 \pm 3.90$ | $93.90 \pm 3.18$ | $75.72 \pm 3.59$ | $74.18 \pm 3.29$ |
| TENT (Reg) | $89.35 \pm 4.49$ | $90.90 \pm 4.24$ | $82.93 \pm 2.02$ | $84.36 \pm 3.49$ |
| X-FNC (Reg) | $90.95 \pm 4.29$ | $92.03 \pm 4.14$ | $82.93 \pm 2.20$ | $84.36 \pm 3.49$ |
| TLP (Reg) | $90.35 \pm 4.49$ | $90.90 \pm 4.24$ | $82.30 \pm 2.05$ | $78.56 \pm 4.42$ |
| Meta-GPS (Reg) | $90.16 \pm 2.72$ | $92.39 \pm 1.66$ | $81.39 \pm 2.35$ | $83.66 \pm 1.79$ |
| COSMIC (Con) | $89.35 \pm 4.49$ | $93.32 \pm 1.92$ | $78.38 \pm 5.21$ | $85.47 \pm 1.42$ |
| TEG (Reg) | $90.14 \pm 0.97$ | $90.18 \pm 0.95$ | $79.42 \pm 1.34$ | $83.27 \pm 0.81$ |
| TEG+MoEFL | $\mathbf{96.19 \pm 0.062}$ | $\mathbf{96.50 \pm 0.062}$ | $\mathbf{88.40 \pm 0.064}$ | $\mathbf{88.80 \pm 0.058}$ |
| STAR (Con) | $95.20 \pm 1.36$ | $96.43 \pm 0.47$ | $88.08 \pm 0.26$ | $90.59 \pm 0.035$ |
| STAR+MoEFL | $\mathbf{97.33 \pm 0.011}$ | $\mathbf{97.03 \pm 0.0067}$ | $\mathbf{90.54 \pm 0.023}$ | $\mathbf{91.69 \pm 0.0090}$ |

of-the-art backbones, STAR (Liu et al., 2025) and TEG (Kim et al., 2023). TEG follows the conventional meta-learning framework, while STAR integrates contrastive learning into the meta-learning process to further enhance few-shot performance. For completeness, we also report the performance of several representative baselines, including G-Meta (Huang & Zitnik, 2021), TENT (Wang et al., 2022), X-FNC (Wang et al., 2023a), TLP (Tan et al., 2022), COSMIC (Wang et al., 2023b), and Meta-GPS (Liu et al., 2024). The abbreviation **NWKS** repersents $N$-way-$K$-shot. For example, 2W3S represents 2-way-3-shot. And **"Reg"** mean regular graph FSL model. And **"Con"** mean constrasive graph FSL model.

## 4.3 Main Result

In table 1 and table 2, we report the results on four small graph datasets. From the table, we can observe that applying our proposed method consistently boosts the performance of both backbones. When the backbone model is STAR, under the 2-way 3-shot setting, we achieve a 1.76% improvement on Cora, a 5.46% improvement on Cora-ML, a 1.13% improvement on WikiCS, and a 3.70% improvement on CiteSeer. Under the 2-way 5-shot setting, we further observe a 1.03% improvement on Cora, a 1.03% improvement on Cora-ML, and a 0.77% improvement on WikiCS. These results demonstrate that our proposed framework provides consistent benefits across different datasets and few-shot learning settings. We see a similar performance improvement when the backbone is TEG.

Furthermore, we also test our framework on larger graph datasets, including "CoraFull" and "coauthor-cs", the results are shown in 3 and 4. The result also show that our proposed framework can enhance both backbone network performance constantly in all 8 settings of 2 datasets on both backbones.

These consistent improvements demonstrate the effectiveness of our proposed framework. By extracting motifs, we incorporate additional structural information from the original graph that is difficult for traditional GNNs to capture. The use of cluster labels and TF-IDF edge weights further regulates the number of motifs and assigns appropriate edge weights, enabling our framework to

seamlessly integrate with various graph few-shot learning backbones. Notably, our framework also enhances stability, as evidenced by the low variance across test tasks, indicating more consistent performance in different FSL scenarios.

## 4.4 ABLATION STUDY

### 4.4.1 VIRTUAL NODE, CLUSTER LABELING AND TF-IDF

In this work, we propose a novel framework composed of three main components. The first is a motif virtual node, which integrates motif information into the graph few-shot learning model. The second is cluster labeling, which assigns temporary labels to nodes to provide enriched supervision signals. The third is a TF-IDF–based virtual edge weighting, which controls the weights of the added edges. In this section, we analyze the effect of each individual component to the overall framework. To accomplish this, we use "STAR" as our backbone and test on three dataset: "Cora", "CiteSeer".

The results are reported in Table 5. We observe that the three components are complementary to each other. In all four settings, using only the Motif Virtual Node leads to worse performance than the baseline. This is mainly due to the limited number of motifs available. For example, with 3-node motifs, there are only two possible patterns, resulting in at most two virtual nodes. Each virtual node then connects to a large number of original nodes, creating a skewed structure that severely degrades performance.

Table 5: Ablation study results on Cora. The backbone is "STAR".

| Motif VN | Cluster Label | TF-IDF | 2W3S | 2W5S |
|:---:|:---:|:---:|:---:|:---:|
| ✗ | ✗ | ✗ | $86.37 \pm 2.13$ | $88.87 \pm 1.68$ |
| ✓ | ✗ | ✗ | $85.66 \pm 0.08$ | $86.86 \pm 0.05$ |
| ✓ | ✓ | ✗ | $88.06 \pm 0.02$ | $88.26 \pm 0.05$ |
| ✓ | ✓ | ✓ | $\mathbf{88.13 \pm 0.04}$ | $\mathbf{89.90 \pm 0.05}$ |

When we introduce Cluster Labeling, the motif patterns become significantly enriched, and node feature information is incorporated into the motif extraction process. This results in a substantial performance improvement compared with using only the Motif Virtual Node.

However, the addition of motifs also introduces many extra edges, which can negatively affect few-shot learning. By further introducing the TF-IDF–based edge weighting, we effectively regulate the relevance between motif nodes and original nodes. This weighting improves the compatibility between virtual nodes and the original graph structure. With all three components combined, our framework achieves the best performance.

Table 6: Comparision with different node number motif. The backbone is "STAR". The dataset using is "Cora". The cluster number is 6.

| Motif Setting | Motif Extracted | Cora 2W3S | Cora 2W5S |
|:---:|:---:|:---:|:---:|
| 3 Node w/o Label | 1 | $85.66 \pm 0.08$ | $86.86 \pm 0.05$ |
| 3/4 node w/o Label | 7 | $86.60 \pm 0.05$ | $88.70 \pm 0.01$ |
| Baseline | - | $86.37 \pm 2.13$ | $88.87 \pm 1.68$ |
| 3 Node with Label | 55 | $\mathbf{88.13 \pm 0.04}$ | $\mathbf{89.90 \pm 0.05}$ |
| 3/4 node with Label | 1760 | $84.63 \pm 0.04$ | $73.35 \pm 0.04$ |

### 4.4.2 CLUSTER LABELING VS INCREASING NODE NUMBER OF MOTIF

In this paper, we introduce a cluster labeling to enrich the extracted motif pattern. A more straightforward way is to increasing the node number of the Motif. In this section, we will compare the performance of both methods. The backbone we use in this section is "Cora" and the dataset we used is "Cora".

The results are reported in Table 6. From the table, it is evident that our proposed cluster labeling outperforms the structure-only motif approach. Without cluster labeling, we are only able to extract

a single pattern from the Cora dataset with 3-node motifs and six additional patterns with 4-node motifs. This limited number of extracted patterns significantly hurts model performance, and both settings without labeling underperform compared to the baseline. In contrast, our proposed method enables the extraction of richer patterns from the graph, which effectively supports the few-shot learning model and leads to better overall performance.

Another observation from the results is that considering only 3-node motifs is actually a favorable choice. As mentioned in the previous section, the number of base 3-node motif patterns is just two, which makes it easier to control the number of motifs extracted from the graph. This claim is also supported by our experiments. When 4-node motifs are included, the number of extracted motifs increases dramatically to 1,760, more than half of the original graph size. Such a large number of motifs heavily interferes with the message-passing process in the original graph, which is particularly evident in the 2-way-5-shot setting, where the accuracy drops by nearly 10% compared with the baseline.

Table 7: Comparision with different cluster number. The backbone is "STAR". The dataset using is "Cora".

| Cluster Number | Cora 2W3S | Cora 2W5S |
|---|---|---|
| 2 | $86.46 \pm 0.04$ | $85.93 \pm 0.02$ |
| 4 | $87.26 \pm 0.02$ | $86.76 \pm 0.03$ |
| 6 | $\mathbf{88.13 \pm 0.04}$ | $\mathbf{89.90 \pm 0.05}$ |
| 8 | $85.43 \pm 0.04$ | $89.20 \pm 0.01$ |
| 10 | $84.63 \pm 0.02$ | $84.70 \pm 0.04$ |

## 4.5 HYPERPARAMETER SELECTION: CLUSTER NUMBER

In this section, we analyze the effect of different cluster numbers on model performance. Specifically, we aim to answer the following question: *for a given dataset, is there an optimal number of clusters?* To investigate this, we conduct experiments on the Cora dataset using STAR as the backbone.

The results is reported in table 7 show that there does indeed exist an optimal cluster number for each dataset. For Cora, setting the number of clusters to 6 achieves the best performance. This is because the cluster number directly determines the number of final virtual nodes. Too few clusters fail to provide sufficient structural diversity, while too many clusters introduce noise, and in both cases, the performance degrades.

## 5 CONCLUSION

In this work, we propose a novel motif-embedding framework, MoEFL, to enhance the performance of graph few-shot learning models. We extract motifs from the original graph and represent each as a virtual node, with a learnable embedding. These virtual nodes are connected to the original graph nodes that belong to the corresponding motif. To enrich the extracted motif patterns, we introduce cluster labeling. Furthermore, we leverage TF-IDF scores to quantify the relevance between motifs and original nodes. Experimental results demonstrate that our proposed framework effectively improves the performance of graph few-shot learning models.

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

## 6   LLM-USAGE

In this paper, we use LLM to polish our paper. The LLM is not involved in experiments.

# A APPENDIX

## A.1 EFFICIENCY EVALUATION

In this section, we evaluate the efficiency of our proposed approach. The method introduces two sources of computational overhead: (1) motif extraction and (2) the additional virtual nodes and edges. Therefore, we assess the impact of each component separately.

First, we analyze the overhead introduced by motif extraction. Our pipeline employs the Weisfeiler–Lehman (WL) graph hashing scheme Shervashidze et al. (2011) to generate unique identifiers for isomorphic subgraphs. This procedure efficiently distinguishes structural patterns without requiring expensive pairwise isomorphism checks, ensuring that the motif extraction stage remains lightweight.

To quantify the associated cost, we use the Coauthor-CS dataset as a reference, which contains **18,333** nodes and **81,894** edges. We measure the additional nodes and edges generated under different clustering configurations. The results are summarized in Table 8.

Table 8: Additional nodes and edges generated when applying MoEFL. The cluster number corresponds to the K-means configuration, and the "Threshold" indicates the percentage of edges removed during motif filtering.

| Setting | Cluster Number/Threshold | Additional Nodes | Additional Edges | Extraction Time (s) |
|---|---|---|---|---|
| 2W3S | 8 + 40% | 116 (0.63%) | 9,256 (11.30%) | 35.36 |
| 2W5S | 10 + 35% | 191 (1.04%) | 14,801 (18.07%) | 36.05 |
| 5W3S | 13 + 35% | 334 (1.82%) | 18,380 (22.04%) | 36.50 |
| 5W5S | 15 + 35% | 471 (2.57%) | 26,464 (32.32%) | 37.20 |

From the table, we observe that the overhead remains small relative to the original graph. As shown in Table 8, the number of additional nodes ranges from 116 to 471 (only 0.6–2.6% of the original), and the number of extra edges remains moderate, with the largest increase (26,464 edges) representing just 32% of the original edge count. The extraction process is also efficient, with all configurations completing within 35–37 seconds. These results demonstrate that MoEFL introduces only lightweight structural modifications while providing significant representational benefits.

In the second part, we examine the additional overhead introduced by motif augmentation during the model training phase. To quantify this cost, we compare the computational resources required by the baseline STAR model and its MoEFL-augmented counterpart under the same GPU setting (A100), using the 5-way 5-shot configuration on the Coauthor-CS dataset. The table below summarizes the differences in per-epoch runtime and GPU memory usage, with all training times reported in seconds per epoch.

Table 9: Computational overhead of integrating MoEFL, measured by per-epoch runtime and GPU memory usage on coauthor-cs under the 5-way 5-shot setting.

| Parameter | Baseline | Baseline + MoEFL |
|---|---|---|
| Running time per epoch(s) | 42 | 60 |
| Memory (GB) | 22 | 26 |

From the table, we see that the overhead introduced by MoEFL during training remains modest. Under the same GPU configuration (A100) and the 5-way 5-shot Coauthor-CS setting, MoEFL increases per-epoch runtime from 42 to 60 seconds (an additional 18 seconds) and raises GPU memory usage from 22 GB to 26 GB (an extra 4 GB). These increments are small relative to the representational gains MoEFL provides. Overall, the results indicate that MoEFL introduces only manageable computational and memory overhead, making it practical to integrate into existing few-shot learning frameworks.

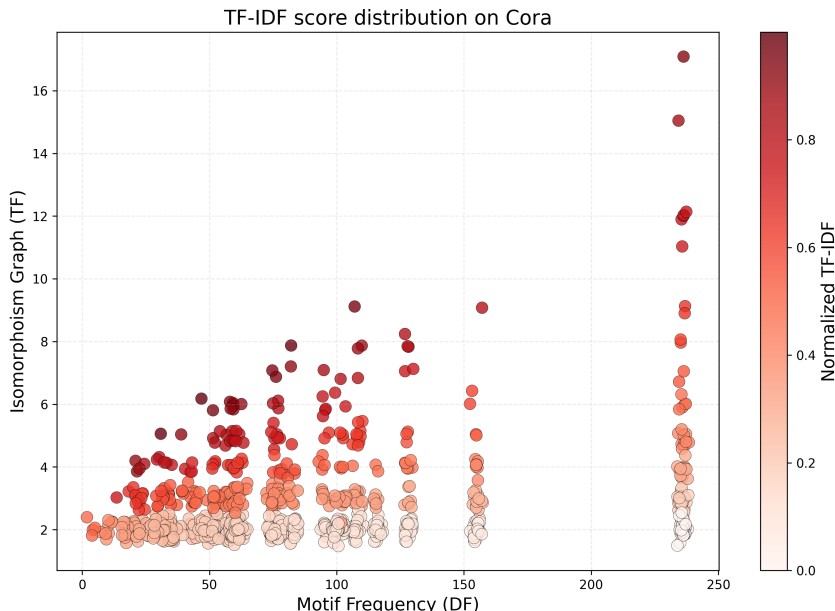

Figure 3: TF-IDF score distribution of the TF-IDF socre based on the number of isomorphism subgraph and motif frequency

## A.2 TF-IDF VISUALIZATION

In this work, we propose using TF-IDF for graphs as virtual edge weights. In this section, we visualize the TF-IDF values to examine how effectively they capture the relationship between motifs and original nodes. We plot the TF-IDF scores derived from both the TF and IDF components. The resulting visualization is shown in the Figure 3.

From the figure, we can observe that rare motifs that appear frequently within specific nodes (low DF, higher TF) receive high TF-IDF values, while globally common motifs are consistently down-weighted. This pattern shows that the TF-IDF encoder correctly captures the intended motif relationship by emphasizing locally informative motifs and suppressing ubiquitous structural patterns.

