# OpenReview forum: "Motif-Aware Embeddings for Enhanced Few-Shot Graph Learning"
_ICLR.cc/2026/Conference — Submitted to ICLR 2026_

### Official Review · Reviewer_6NJv · 2025-10-28

**Soundness:** 3
**Presentation:** 4
**Contribution:** 3
**Rating:** 4
**Confidence:** 4

**Summary:**

This paper addresses motif graph-based few shot learning and proposes a novel motif extraction and meta edge construction framework called MoEFL. The proposed MoEFL framework extracts motifs from graphs while preserving label information, and employs a tf-idf–based encoding scheme to reduce unnecessary connections between virtual and actual nodes. Extensive experimental results demonstrate the effectiveness of the proposed approach.

**Strengths:**

1. It is an interesting idea to introduce tf-idf as a simple yet effective motif embedding method from NLP field.
2. The method is sufficiently motivated, the paper is generally well structured and easy to follow.
3. The MoEFL framework can be integrated into existing motif-based few-shot learning models and bring improvements.

**Weaknesses:**

1. Although the proposed MoEFL method is relatively straightforward, the paper lacks sufficient implementation details and does not provide publicly accessible code, which raises concerns about reproducibility.
2. Additional case studies and visualizations are needed to illustrate how the TF-IDF encoder represents motifs in practical scenarios.
3. Since MoEFL introduces additional virtual nodes and edges, analyses of the resulting spatial and temporal complexity should be included to better understand its computational impact.

**Questions:**

Please refer to the weaknesses above. My primary concern lies with the reproducibility of the experiments. I would be willing to raise my score if the authors can address these concerns or provide further implementation details.

---

> ### Author Response · Authors · 2025-11-19
> **Author response for Reviewer 6NJv(1)**
>
> **Question 1 Although the proposed MoEFL method is relatively straightforward, the paper lacks sufficient implementation details and does not provide publicly accessible code, which raises concerns about reproducibility.**
>
> **Answer:** Thank you for pointing this out. We have included the reference code in the supplementary material.
>
> As an example, for the Coauthor-CS dataset, the clustering numbers and TF-IDF thresholds are treated as predefined hyperparameters. The following values illustrate the specific choices we adopt for this dataset.
> | Setting | Cluster#  | Threshold |
> |:-------:|:---------:|:----------:|
> | 2W3S    | 8         | 40%        |
> | 2W5S    | 10        | 35%        |
> | 5W3S    | 13        | 35%        |
> | 5W5S    | 15        | 35%        |
>
> The virtual-node embedding dimension is set equal to the standard node-embedding dimension used in each baseline.
>
> **Question 2 Additional case studies and visualizations are needed to illustrate how the TF-IDF encoder represents motifs in practical scenarios.**
>
> **Answer:** Thank you for raising this question. This is indeed an important aspect for demonstrating the effectiveness of our proposed TF-IDF–based motif encoder. To address this, we visualize the distribution of normalized TF-IDF scores on the Cora dataset and present the corresponding scatter plot. The figure is in the appendix part **A.2** of revised manuscript.
>
> This figure visualizes the relationship between motif document frequency (DF), isomorphism-
> subgraph counts (TF), and the resulting TF-IDF scores. From the figure, we can observe that rare
> motifs that appear frequently within specific nodes (low DF, higher TF) receive high TF-IDF values,
> while globally common motifs are consistently down-weighted. This pattern shows that the TF-
> IDF encoder correctly captures the intended motif relationship by emphasizing locally informative
> motifs and suppressing ubiquitous structural patterns.

---

> ### Author Response · Authors · 2025-11-19
> **Author response for Reviewer 6NJv(2)**
>
> **Question 3: Since MoEFL introduces additional virtual nodes and edges, analyses of the resulting spatial and temporal complexity should be included to better understand its computational impact.**
>
> **Answer:** This is also an important aspect for evaluating the effectiveness of our method. We examine this issue from two perspectives.
>
> First, we analyze the overhead introduced by motif extraction. In our framework, we adopt the Weisfeiler–Lehman (WL) graph hashing scheme[1] to generate unique identifiers for isomorphic subgraphs. Using the Coauthor-CS dataset as a reference, which contains **18,333** nodes and **81,894** edges, we measure the additional nodes and edges generated under different clustering configurations. The corresponding results are summarized in the table below.
> | Setting | Cluster Number / Threshold | Additional Nodes | Additional Edges | Extraction Time (s) |
> |---------|-----------------------------|-------------------|-------------------|----------------------|
> | 2W3S    | 8 + 40%                     | 116 (0.63%)       | 9256 (11.30%)     | 35.36               |
> | 2W5S    | 10 + 35%                    | 191 (1.04%)       | 14801 (18.07%)    | 36.05               |
> | 5W3S    | 13 + 35%                    | 334 (1.82%)       | 18380 (22.04%)    | 36.50               |
> | 5W5S    | 15 + 35%                    | 471 (2.57%)       | 26464 (32.32%)    | 37.20               |
>
> Although MoEFL introduces extra nodes and edges through cluster-level virtual nodes and TF-IDF–guided motif augmentation, the overhead remains small relative to the original graph. As shown in Table above, the number of additional nodes ranges from 116 to 471 (only 0.6–2.6\% of the original), and the number of extra edges remains moderate, with the largest increase (26,464 edges) representing just 32\% of the original edge count. The extraction process is also efficient, with all configurations completing within 35–37 seconds. These results demonstrate that MoEFL introduces only lightweight structural modifications while providing significant representational benefits.
>
> In the second part, we evaluate the additional overhead introduced by motif augmentation during the model training phase. To quantify this cost, we compare the computational resources required by the baseline STAR model and its MoEFL-augmented version under the same GPU setting (A100), using the 5-way 5-shot configuration on the Coauthor-CS dataset. The table below summarizes the differences in per-epoch runtime and GPU memory consumption, with all training times reported in seconds per epoch.
> | **Parameter**              | **Baseline** | **Baseline + MoEFL** |
> |---------------------------|--------------|------------------------|
> | Running time per epoch (s) | 42           | 60                     |
> | Memory (GB)             | 22           | 26                     |
>
> From the table, we see that the overhead introduced by MoEFL during the training phase remains moderate and well within practical limits. Under the same GPU configuration (A100) and using the 5-way 5-shot setting on the Coauthor-CS dataset, MoEFL increases the per-epoch training time from 42 to 60 seconds, which is an additional 18 seconds of computation. The GPU memory usage also rises from 22 GB to 26 GB, adding only 4 GB of extra consumption. These increases are relatively small given the representational benefits provided by MoEFL. Overall, the results show that integrating MoEFL introduces only manageable computational and memory overhead.
>
> Reference [1]:
> Nino Shervashidze, Pascal Schweitzer, Erik Jan van Leeuwen, Kurt Mehlhorn, and Karsten M. Borg- wardt. Weisfeiler-lehman graph kernels. J. Mach. Learn. Res., 12(null):2539–2561, November 2011. ISSN 1532-4435.

---

### Official Review · Reviewer_iohD · 2025-10-30

**Soundness:** 2
**Presentation:** 2
**Contribution:** 1
**Rating:** 4
**Confidence:** 3

**Summary:**

The paper proposes a motif-aware embedding framework (MoEFL) for graph few-shot learning that plugs into existing backbones (e.g., STAR, TEG) to inject structural cues from the graph. The pipeline clusters nodes by feature similarity to assign temporary labels, extracts motif patterns (mainly 3-node, optionally 4-node) under these labels, creates a learnable “virtual node” for each motif pattern, connects it to participating original nodes, and weights each virtual edge using a TF-IDF–style score; a two-layer GNN with a gating mechanism then fuses motif embeddings with node features before the few-shot head. The authors claim clustering enriches motif types and TF-IDF mitigates degree explosion, positioning the framework as a general add-on. Experiments on small (Cora, Cora-ML, WikiCS, CiteSeer) and larger (CoraFull, Coauthor-CS) benchmarks report consistent accuracy gains over the backbones and lower variance, while ablations on Cora show that the motif virtual node alone hurts performance but adding cluster labels and then TF-IDF recovers and improves it; scaling to labeled 3/4-node motifs dramatically increases pattern count (e.g., 1,760) and degrades results.

**Strengths:**

- This paper focuses on graph few shot learning, which is relatively a novel question in current graph learning research.
- The method is designed to integrate with various graph FSL backbones, and the authors report lower variance across tasks and consistent gains over different methods.
- By using motifs plus cluster labeling, the approach injects structural information, does not rely on ground-truth labels.

**Weaknesses:**

- The novelty of the proposed method is limited as the 3-node motif extraction in the framework explicitly follows a previous method (Chen et al., 2023).

2. Scalability concerns with no cost analysis: The construction can create tens of thousands of virtual edges.

3. Experimental Reproducibility: Some key experimental details are missing or buried: how are validation splits constructed? Are hyperparameters for TF-IDF thresholds, clustering, and virtual node embedding dimensions set in advance or tuned? The paper does not state the computational cost (run-time, memory) compared to baselines. This impairs both transparency and reproducibility.

**Questions:**

See weaknesses.

---

> ### Author Response · Authors · 2025-11-19
> **Author Response to Reviewer iohD(1)**
>
> **Question 1: The novelty of the proposed method is limited as the 3-node motif extraction in the framework explicitly follows a previous method.Chen et al. (2023)[1]**
>
> **Answer:** We would like to clarify that our method differs fundamentally from Chen et al. (2023). Their 3-node motifs are derived solely from structural connectivity and therefore cannot capture feature-level relationships. Our approach explicitly integrates node features via cluster-based pseudo-labeling and incorporates these labels during motif extraction. This yields motifs that encode both structure and semantics. Therefore, our method is not simply adopting their motif definitions but meaningfully extends them to a feature-aware setting.
>
> For a clear view, we summarize the difference between [1]  and our work in following table.
> | **Aspect** | **Chen et al. (2023)** | **Our Method (MoEFL)** |
> |------------|------------------------|--------------------------|
> | **Motif Type** | 3-node structural motifs (13 directed patterns) | Feature-aware motifs combining structural patterns with node attribute information |
> | **Feature Integration** | Not supported; motifs are purely topological | Cluster-based pseudo-labeling incorporates feature similarity directly into motif extraction |
> | **Motif Usage** | Create  a motif-aware adjacency matrix for each motif; limited to a fixed set of 13 motifs | Learns motif embeddings with virtual nodes; supports flexible motif types |
>
> Reference:
> [1]: Xuexin Chen, Ruichu Cai, Yuan Fang, Min Wu, Zijian Li, and Zhifeng Hao. Motif graph neural
> network. IEEE Transactions on Neural Networks and Learning Systems, 35(10):14833–14847,
> 2023

---

> ### Author Response · Authors · 2025-11-19
> **Author Response to Reviewer iohD(2)**
>
> **Question 2: Scalability concerns with no cost analysis: The construction can create tens of thousands of virtual edges.**
>
> **Answer:** Thanks for pointing this out — it is indeed an important aspect, and we highlight it in this section. In our framework, we adopt the Weisfeiler–Lehman (WL) graph hashing scheme[1] to generate unique identifiers for isomorphic subgraphs. To evaluate the overhead introduced by MoEFL, we use the coauthor-cs dataset as a reference. The original graph contains **18,333** nodes and **81,894** edges. We report the additional nodes and edges created under different clustering configurations, as shown in following table.
> | Setting | Cluster Number / Threshold | Additional Nodes | Additional Edges | Extraction Time (s) |
> |---------|-----------------------------|-------------------|-------------------|----------------------|
> | 2W3S    | 8 + 40%                     | 116 (0.63%)       | 9256 (11.30%)     | 35.36               |
> | 2W5S    | 10 + 35%                    | 191 (1.04%)       | 14801 (18.07%)    | 36.05               |
> | 5W3S    | 13 + 35%                    | 334 (1.82%)       | 18380 (22.04%)    | 36.50               |
> | 5W5S    | 15 + 35%                    | 471 (2.57%)       | 26464 (32.32%)    | 37.20               |
>
> Although MoEFL adds extra nodes and edges through cluster-level virtual nodes and TF-IDF–based motif augmentation, the overhead is negligible compared to the size of the original graph (18,333 nodes and 81,894 edges). As summarized in Table above, the number of additional nodes is minimal—only 191 to 471, corresponding to just 1–2.6% of the original node count—and the number of added edges remains modest, with the largest increase (26,464 edges) representing only about 32% of the original graph. The preprocessing time is also low, consistently falling between 35 and 37 seconds. These results show that MoEFL imposes only lightweight structural modifications while delivering substantial representational improvements. Notably, this motif extraction process only need to conduct once. The motif information can be saved in json file.
>
> Reference
> [1]: Nino Shervashidze, Pascal Schweitzer, Erik Jan van Leeuwen, Kurt Mehlhorn, and Karsten M. Borg- wardt. Weisfeiler-lehman graph kernels. J. Mach. Learn. Res., 12(null):2539–2561, November 2011. ISSN 1532-4435.

---

> ### Author Response · Authors · 2025-11-19
> **Author Response to Reviewer iohD(3)**
>
> **Question 3: Experimental Reproducibility: Some key experimental details are missing or buried: how are validation splits constructed? Are hyperparameters for TF-IDF thresholds, clustering, and virtual node embedding dimensions set in advance or tuned? The paper does not state the computational cost (run-time, memory) compared to baselines. This impairs both transparency and reproducibility.**
>
> **Answer:** Thank you for this valuable question. We appreciate the opportunity to clarify the experimental setup and improve reproducibility. Below we provide additional details regarding hyperparameter choices, validation splits, and computational settings.
>
> First, the clustering numbers and TF-IDF thresholds are predefined as hyerparameter rather than tuned. For example, for coauthor-cs dataset we adopt the following clustering numbers and TF-IDF thresholds.
> | Setting | Cluster#  | Threshold |
> |:-------:|:---------:|:----------:|
> | 2W3S    | 8         | 40%        |
> | 2W5S    | 10        | 35%        |
> | 5W3S    | 13        | 35%        |
> | 5W5S    | 15        | 35%        |
>
> The virtual-node embedding dimension is set equal to the standard node-embedding dimension used in each baseline to ensure
> architectural compatibility. All other experiment setting will remain identical to the baseline settings to
> provide a fair comparison.
>
> To evaluate the overhead introduced by MoEFL, we compare computation resources under the same GPU setting (A100) using the 5-way 5-shot configuration on the coauthor-cs dataset. The table below reports the additional runtime and memory consumption relative to the baseline STAR model. All running times are measured in seconds per epoch.
> | **Parameter**              | **Baseline** | **Baseline + MoEFL** |
> |---------------------------|--------------|------------------------|
> | Running time per epoch (s) | 42           | 60                     |
> | Memory (GB)             | 22           | 26                     |
>
> From this table, we can see that the overhead introduced by MoEFL is quite acceptable. Compared with the baseline, MoEFL increases the per-epoch training time from 42 seconds to 60 seconds, adding only 18 seconds of additional computation. Similarly, the GPU memory usage increases from 22 GB to 26 GB, resulting in an extra 4 GB of memory consumption. Overall, these results demonstrate that incorporating MoEFL incurs only moderate computational and memory overhead, making it practical to integrate into existing few-shot learning frameworks without significant resource constraints.

---

### Official Review · Reviewer_1HZK · 2025-11-01

**Soundness:** 2
**Presentation:** 3
**Contribution:** 2
**Rating:** 4
**Confidence:** 2

**Summary:**

MoEFL improves few-shot node classification by adding motif-based virtual nodes enriched with cluster labels, and uses TF-IDF weights to control their influence.

**Strengths:**

1. Simple and generally applicable: Can be plugged into many few-shot graph learning models.
2. Motif + cluster labeling provides richer structural signals than topology alone.

**Weaknesses:**

1. It's not interesting to explore only node classification task, since this has been widely explored.
2.Extraction of motifs may be computationally expensive for large graphs.

**Questions:**

See weaknesses.

---

> ### Author Response · Authors · 2025-11-19
> **Author Response to Reviewer 1HZK(1)**
>
> **Question 1: It's not interesting to explore only node classification task, since this has been widely explored**
>
> **Answer:** Our method is designed to be model-agnostic and can be integrated with a wide range of graph few-shot learning frameworks—not only node classification. In this paper, we evaluate our approach on two state-of-the-art node classification baselines to demonstrate its effectiveness. However, the proposed motif-enhanced TF-IDF encoder and virtual-node augmentation can be directly applied to other few-shot settings as well.
>
> To illustrate this generality, we take the few-shot graph classification task as an example and adapt our method on top of the baseline model called $\textbf{Faith}$ in [1]. For graph classification, we first collect all nodes from every graph in the dataset and assign cluster labels based on their global structural distribution. We then extract motifs across all graphs to build a unified global motif embedding space. Each individual graph connects only to the virtual nodes corresponding to its own cluster and motif types, allowing the model to leverage global motif patterns while preserving task-specific graph structure.
>
> The result is show in the following table:
> | Method   | Letter-High 5-shot | Letter-High 10-shot | EMZYMES 5-shot | EMZYMES 10-shot | Reddit-12K 5-shot | Reddit-12K 10-shot |
> |----------|---------------------|----------------------|------------------|-------------------|---------------------|----------------------|
> | Baseline | 71.55 ± 3.58        | 76.65 ± 3.26         | 57.89 ± 4.65     | 62.16 ± 4.11      | 42.71 ± 4.18        | 46.63 ± 4.01         |
> | MoEFL    | **76.18 ± 0.18**    | **77.50 ± 0.17**     | **91.57 ± 0.10** | **90.56 ± 0.08**  | **53.91 ± 0.248**   | **65.93 ± 0.22**     |
>
> From the table above, we observe that incorporating our proposed MoEFL module consistently improves few-shot graph classification accuracy across all three datasets and both 5-shot and 10-shot settings. The gains are substantial: compared to the baseline, MoEFL increases performance by 4–5% on Letter-High, delivers dramatic improvements of over 30 % on EMZYMES, and achieves notable boosts on Reddit-12K as well. These results demonstrate that our method not only enhances node-level few-shot learning but also generalizes effectively to graph-level few-shot tasks, confirming the broad applicability and versatility of the proposed framework.
>
> Reference:
>
> [1]: Song Wang, Yushun Dong, Xiao Huang, Chen Chen, and Jundong Li. Faith: Few-shot graph classi-
> fication with hierarchical task graphs. In IJCAI, 2022b.

---

> ### Author Response · Authors · 2025-11-19
> **Author Response to Reviewer 1HZK (2)**
>
> **Question 2: Extraction of motifs may be computationally expensive for large graphs.**
>
> **Answer:** Thank you for highlighting this point. The efficiency of motif extraction is indeed an important factor in evaluating our method. In our framework, we adopt the Weisfeiler–Lehman (WL) graph hashing scheme[1] to generate unique identifiers for isomorphic subgraphs. This process efficiently distinguishes structural patterns without performing expensive pairwise isomorphism checks, ensuring that motif extraction remains lightweight.
>
> To further quantify the overhead introduced by MoEFL, we use the Coauthor-CS dataset as a reference. The original graph contains **18,333** nodes and **81,894** edges. We report in the table below the number of additional nodes and edges produced under different clustering configurations, which provides a clear view of the structural augmentation cost introduced by our approach.
>
> | Setting | Cluster Number / Threshold | Additional Nodes | Additional Edges | Extraction Time (s) |
> |---------|-----------------------------|-------------------|-------------------|----------------------|
> | 2W3S    | 8 + 40%                     | 116 (0.63%)       | 9256 (11.30%)     | 35.36               |
> | 2W5S    | 10 + 35%                    | 191 (1.04%)       | 14801 (18.07%)    | 36.05               |
> | 5W3S    | 13 + 35%                    | 334 (1.82%)       | 18380 (22.04%)    | 36.50               |
> | 5W5S    | 15 + 35%                    | 471 (2.57%)       | 26464 (32.32%)    | 37.20               |
>
> MoEFL introduces additional nodes and edges through cluster-level virtual nodes and TF-IDF–filtered motif augmentation, but the overall overhead remains minimal relative to the original graph, which contains 18,333 nodes and 81,894 edges. As shown in Table above, the number of added nodes is very small—ranging from 191 to 471 (approximately 1–2.6% of the original count)—and the added edges remain modest, with a maximum of 26,464 (about 32% of the original edges). The preprocessing cost is also low, with extraction times consistently around 35–37 seconds, demonstrating that MoEFL enhances the graph structure with negligible computational burden. Overall, MoEFL provides lightweight structural augmentation while delivering significant representational gains. Notably, motif extraction is performed only once, and all motif information can be cached in a JSON file for future use.
>
> Reference
>
> [1]: Nino Shervashidze, Pascal Schweitzer, Erik Jan van Leeuwen, Kurt Mehlhorn, and Karsten M. Borg-
> wardt. Weisfeiler-lehman graph kernels. J. Mach. Learn. Res., 12(null):2539–2561, November
> 2011. ISSN 1532-4435.

---

### Author Response · Authors · 2025-12-02
**Summary for Rebuttal**

We are grateful to all reviewers for their valuable and thoughtful comments. In the following, we present a summary of the revisions and clarifications made in response to their concerns.

1. All three reviewers raised questions regarding the efficiency of our approach. In response, we conducted additional experiments demonstrating that:

   1. The motif extraction step is a one-time preprocessing procedure and can be executed very efficiently.
   2. Even with the added nodes and features, our proposed framework introduces only a modest and manageable overhead compared with the baseline.


2. Reviewers also expressed concerns regarding the reproducibility of our results. In the rebuttal, we provide anonymized code and detailed experimental settings to facilitate full reproducibility.

3. Reviewer **1HZK** raised the concern that our method appears applicable only to node-classification tasks. In fact, our approach can be readily extended to other graph few-shot learning settings. To illustrate this, we include additional experiments on \textbf{few-shot graph classification tasks} following the baseline setup in [1]. The results show that our method consistently improves the performance of the baseline, demonstrating its broader applicability.

4. Reviewer **ioHD** raised concerns regarding the novelty of our motif extraction process. In response, we provide a detailed illustration clarifying the differences between our approach and the original method in [2]. Specifically, our method incorporates node features as pseudo-labels to enrich the extracted motifs, whereas the original process relies solely on structural information. This enhancement enables our motifs to capture both structural and semantic properties of the graph.

5. Reviewer **6NJv** suggested adding a visualization of the TF-IDF encoder. Accordingly, we include a new figure in Appendix A.2 that illustrates how our TF-IDF module captures meaningful relationships in practical scenarios.

We sincerely thank all reviewers again for their constructive feedback and valuable suggestions. We hope that our detailed responses and additional experiments adequately address all raised concerns.


Reference:

[1]: Song Wang, Yushun Dong, Xiao Huang, Chen Chen, and Jundong Li. Faith: Few-shot graph classi- fication with hierarchical task graphs. In IJCAI, 2022b.

[2]: Xuexin Chen, Ruichu Cai, Yuan Fang, Min Wu, Zijian Li, and Zhifeng Hao. Motif graph neural network. IEEE Transactions on Neural Networks and Learning Systems, 35(10):14833–14847, 2023

---

### Meta-Review · Area_Chair_JYmX · 2026-01-08

**Summary:**

Reviewers noted the following key issues:

- Scalability to large graphs where the bottleneck would be motif extraction.
- Experimental settings/implementation details are not complete
- Similarity to existing motif-based graph learning

**Reviewer Concerns:**

- Scalability: authors showed some evaluation in rebuttals but it is still limited in scale. It is unclear what would be the consequence on much larger graphs (100K, or 1M+ nodes)
- Experimental settings/implementation details: clarified to some extent
- Novelty: while the authors argued the differences, the most important of which is feature integration, the overall motif-based graph learning idea is somewhat incremental. The related work section (2.2) in the revised version remains brief without a clarification on the differences.

**Reviewer Scores:**

Marginal increase (0.5 by average) or no change.

---

### Decision · Program_Chairs · 2026-01-26

Reject